# Chronic Thromboembolic Pulmonary Hypertension: An Observational Study

**DOI:** 10.3390/medicina58081094

**Published:** 2022-08-13

**Authors:** Barbara Ruaro, Paola Confalonieri, Gaetano Caforio, Elisa Baratella, Riccardo Pozzan, Stefano Tavano, Chiara Bozzi, Selene Lerda, Pietro Geri, Marco Biolo, Maurizio Cortale, Marco Confalonieri, Francesco Salton

**Affiliations:** 1Department of Pulmonology, University Hospital of Cattinara, University of Trieste, 34149 Trieste, Italy; 2Department of Radiology, Cattinara Hospital, University of Trieste, 34149 Trieste, Italy; 324ore Business School, Via Monte Rosa, 91, 20149 Milan, Italy; 4Department of Medical, Surgical, & Health Sciences, Cattinara Hospital, University of Trieste, 34127 Trieste, Italy

**Keywords:** chronic thromboembolic pulmonary hypertension (CTEPH), chronic thromboembolism, pulmonary artery pressure (PAP), pulmonary thromboendarterectomy, pulmonary vasodilator therapy

## Abstract

*Background and Objectives:* Chronic thromboembolic pulmonary hypertension (CTEPH) has a high mortality. The treatment of CTEPH could be balloon pulmonary angioplasty (BPA), medical (MT) or pulmonary endarterectomy (PEA). This study aims to assess the clinical characteristics of CTEPH patients, surgically or medically treated, in a pulmonology referral center. *Materials and Methods:* A total of 124 patients with PH with suspected CTEPH (53 male subjects and 71 female subjects; mean age at diagnosis 67 ± 6) were asked to give informed consent and then were evaluated. The presence of CTEPH was ascertained by medical evaluations, radiology and laboratory tests. *Results:* After the evaluation of all clinical data, 65 patients met the inclusion criteria for CTEPH and they were therefore enrolled (22 males and 43 females; mean age at diagnosis was 69 ± 8). 26 CTEPH patients were treated with PEA, 32 with MT and 7 with BPA. There was a statistically significant age difference between the PEA and MT groups, at the time of diagnosis, the PEA patients were younger than the MT patients, whereas there was no statistically significant difference in other clinical characteristics (e.g., smoking habit, thrombophilia predisposition), as well as functional and hemodynamic parameters (e.g., 6-min walk test, right heart catheterization). During three years of follow-up, no patients in the PEA groups died; conversely, eleven patients in the MT group died during the same period (*p* < 0.05). Furthermore, a significant decrease in plasma BNP values and an increase in a meter at the six-minute walk test, 1 and 3 years after surgery, were observed in the PEA group (*p* < 0.05). *Conclusions:* This study seems to confirm that pulmonary endarterectomy (PEA) can provide an improvement in functional tests in CTEPH.

## 1. Introduction

Pulmonary hypertension secondary to chronic thromboembolism (CTEPH) represents a potential long-term complication secondary to one or more thromboembolic events [1,2,3,4,5]. However, the reasons why just a minority of patients with pulmonary thromboembolism (PE) may develop pulmonary hypertension (PH) and eventually right ventricular failure remain to be elucidated [5,6,7,8,9]. Furthermore, the exact pathogenesis of CTEPH is still uncertain [10,11,12,13,14,15,16,17]. Chronic thromboembolic pulmonary vascular disease (CTED) comprehends rare expressions of venous thromboembolism [10,11,12,13,14,15,16,17].

The diagnosis of CTEPH is still challenging, requiring a mean time of 14 months from the start of symptoms to obtain the diagnosis, even in centers of expertise. It requires evidence of precapillary pulmonary hypertension given by the analysis of the right heart catheterization (RHC), in subjects with elements consistent with perfusion impairment of the lung blood flow of thromboembolic etiology, persistent after at least three months of anticoagulant therapy [18,19,20,21,22,23]. It is complex to determine the exact CTEPH incidence because CTEPH is presumably both underdiagnosed and the incidence of CTEPH following acute PE is susceptible to overestimation [6,7,24]. Furthermore, CTEPH epidemiology is clearly impacted by geographic dissimilarities [25,26,27,28,29]. Women constituted one-half of the study population of the European CTEPH registry [30]. Several abnormalities, including autoimmune, inflammatory and thrombophilia markers, have been found in CTEPH patients; it is feasible that variability in this underlying pathological milieu contributes to the variability in the worldwide CTEPH epidemiology [30,31,32,33,34,35,36,37]. Antiphospholipid syndrome (APLS) is one of the most associated with thrombophilia, with a prevalence of CTEPH of 2–50% [38]. An interesting review demonstrated that CTEPH could be another sequel to severe post-COVID-19 pneumonia [39]. Furthermore, variable gene expression has been demonstrated in pulmonary artery endothelial cells from patients with CTEPH compared with normal controls [30,31,32,33,34,35].

CTEPH belongs to the fourth group of the WHO clinical classification of pulmonary hypertension (PH), which comprehends definite etiology, medical presentation, diagnostic tests and therapeutic strategies, and consists of a peculiar lung vascular disease [19,20,21,22,23]. CTEPH also differs from other pulmonary hypertension groups also for its therapeutic possibilities. Pulmonary endarterectomy (PEA) is still the gold standard treatment, particularly in the presence of organized thrombi involving the main lobar or segmental arterial vessels, and it is a challenging surgical treatment that consists of the removal of obstructive thromboembolic material from the pulmonary vessels (Figure 1) [40,41,42,43]. Improving quality and life expectancy can be seen in a high percentage of patients after surgery, which for many of them can therefore be considered a definitive cure [43,44,45,46,47]. For these reasons, operable patients are specifically distinguished from inoperable, in order to offer the former, the possibility of surgical treatment [43,44,45,46,47].

The decision to exclude the surgical option is mainly because of the involvement of the technically inaccessible distal pulmonary arteries, patients with severe comorbidities (disadvantageous risk-benefit ratio) or the subject’s rejection of the surgical treatment [45,46,47,48,49,50,51,52,53]. For patients with CTEPH who have inoperable or persistent pulmonary hypertension after surgical treatment, the current ESC/ERS guidelines for the diagnosis and treatment of pulmonary hypertension recommend the only currently approved drug treatment as follows: soluble guanylate cyclase activator, Riociguat [19,21]. Other drugs for pulmonary hypertension tested in CTEPH are used off-label [53,54,55,56,57,58,59,60,61,62]. Balloon pulmonary angioplasty (BPA) is an emerging and promising treatment option for CTEPH patients who have inoperable, segmental/subsegmental disease, or residual disease after pulmonary endarterectomy [63]. BPA continues to develop while future research is required to demonstrate the long-term benefits of the technique, standardize the method and define a uniform institutional infrastructure for providing BPA as a part of the treatment of CTEPH [63]. The choice to use off-label medications approved for PH and BPA can be taken into consideration in centers of expertise by a multidisciplinary team [53,54,55,56,57,58,59,60,61,62,63].

The primary aim of this study was to evaluate the phenotype of CTEPH patients afferent to the national center of reference of ours for CTEPH. The secondary endpoint was to investigate the characteristics of the CTEPH patients treated with pulmonary endarterectomy (PEA), medical treatment (MT) or balloon pulmonary angioplasty (BPA).

## 2. Materials and Methods

### 2.1. Patient Population

In this retrospective study, after receiving written informed consent, a systematic review of the clinical documentation (e.g., hospitalization documents, discharge reports and examination reports) of 124 suspect or confirmed CTEPH patients afferent to our Pneumology Department was performed between January 2019 and September 2021. Demographic and medical data (i.e., gender, age, smoking habit, treatment regimens) were also recorded (Table 1).

The inclusion criteria were diagnosis of CTEPH according to the latest guidelines [19,21]. The exclusion criteria were patients without sufficient medical data (i.e., age at diagnosis, smoking status) no bloodwork or imaging useful to support diagnosis; PH secondary to other causes, including idiopathic pulmonary hypertension or PH associated with autoimmune disease [64,65,66]. We considered exclusion criteria for lung thromboembolism to be a physiological perfusion pattern at the V/Q lung scan, the absence of organized perfusion defects and the presence of abnormal perfusion of the lung secondary to anomalies in the pulmonary parenchyma. The characteristics relating to age and gender, the risk conditions linked to the onset of CTEPH recognized in the scientific literature and chronic comorbidities were reported at the time of diagnosis. Blood tests (elaborated in Section 2.2), PFTs (elaborated in Section 2.3) and imaging assessment (elaborated in Section 2.4) were carried out for all subjects.

### 2.2. Blood Tests

A complete blood panel evaluation was made at first visit in our usual clinical practice, i.e., a complete blood count, plasma BNP, antiphospholipid antibodies (antibodies anti-beta2-glycoprotein, antibodies anti-cardiolipin, lupus anticoagulant), screening for thrombophilic diseases (protein C deficiency, Factor V Leiden alteration, protein S deficiency, hyperhomocysteinemia/MTHFR, factor II mutation) (Table 2).

### 2.3. Transthoracic Echocardiogram

Standard transthoracic echocardiography (TTE) was performed on all patients to evaluate left and right heart function.

### 2.4. Right Heart Catheterization

Patients must be fully informed of risks related to the procedure. The most common complications are access site hematoma, vagal reaction, pneumothorax and arrhythmias [67]. Where available, individual or departmental complication rates should be quoted. Catheterization is frequently performed without interruption of anticoagulation. Right heart catheterization (RHC) was performed using a balloon-floating catheter named Swan-Ganz. Pressure measurement and RHC-CO were performed following standard techniques. RHC-CO and mPAP were obtained. RHC-CO was determined thanks to the use of the Fick equation.

### 2.5. V/Q Lung Scan

All subjects with a medical or laboratory suspect of pulmonary embolism (PE) were evaluated by V/Q lung scan, i.e., the commonest assessment method for the detection of PE [68,69,70]. V/Q lung scan was carried out after the i.v. injection of albumin macro aggregates labeled with radioactive technetium (^99m^Tc-MAA Makro-Albumon^®^; Medi-Radiopharma Ltd., Érd, Szamos, 2030, Hungary). The lung scintigraphy image acquisition commenced straight after the administration of the radiotracer and the images were obtained by a SPECT/CT hybrid dual-head gamma camera, Infinia VC Hawkeye 4 (GE Healthcare, 3135 Easton Turnpike Chicago, IL 06828, USA). The patient held the same position during the acquisition of all SPECT/CT images, which were interpreted by Xeleris 1 and 2 Functional Imaging Workstations (GE Healthcare, 3135 Easton Turnpike Chicago, IL 06828, USA). The lung scintigraphies were evaluated by 2 expert nuclear medicine specialists (with 10 and 24 years of experience, respectively).

On the basis of the SPECT/CT results, we adopted the following requirement to confirm the diagnosis of lung embolism: at least 1 segmental or 2 sub-segmental perfusion defects (wedge-shaped, base directed towards the pleura) without corresponding anomalies in the lung tissue. CT pulmonary angiography was performed as the first choice for diagnosing acute pulmonary embolism in some patients.

### 2.6. Pulmonary Angio-CT

Pulmonary angio-CT (Philips Healthcare, 3000 Minuteman Road Andover, MA, 01810-1099, USA) may be obtained in subjects with undiagnosed CTEPH or as part of the diagnostic process for patients with suspected CTEPH. CT findings of subjects with CTEPH differ from those with acute lung embolism, in whom intraluminal filling defects are the main radiologic anomalies. Being familiar with CT findings in patients who have CTEPH helps radiology specialists diagnose this disease and contributes to determining the possibility of surgery.

CT technique can point out distinctive abnormalities, comorbidities and complications often seen in CTEPH, e.g., dilatation of pulmonary arteries or large-caliber bronchial collateral arteries, along with hypoperfused areas (hyper-transparency) with the characteristic “mosaic” pattern (Figure 2). During the acquisition of CT images, it is essential to lower to the minimum delivered dose, especially in CTEPH patients since they may need a follow-up. Hence, the tube potential and current, usually set at 120 kVp and ≤240 mAs, should be tailored to the patient’s constitution, reducing the kilovoltage and milliamperage for lighter subjects, e.g., to values of 100 kVp, in addition to the choice of techniques that prevent redundant exposure, e.g., tube current modulation.

Nevertheless, V/Q lung scan continues to be the gold standard for the diagnosis of CTEPH, thanks to its sensitivity of 96–97% and its specificity of 90–95% [63] and it was performed in 95% of our patients.

### 2.7. Pulmonary Function Tests

Pulmonary function tests (PFTs) are fundamental, promptly available and non-invasive tests. They were carried out in the pulmonology units of the University Hospital of Trieste; all the subjects shared the same operator and equipment. Global spirometry values were obtained, i.e., functional vital capacity (FVC), forced expiratory volume in 1s (FEV1) and forced expiratory volume in 1s/functional vital capacity (FEV1/FVC) ratio, also known as the Tiffenau index (TI) and diffusion capacity of carbon monoxide (DLCO).

### 2.8. Six-Minute Walking Test

The six-minute walk test (6MWT) is an indicator of physical function and therapeutic response in patients with chronic lung disease. The test measures how many meters the patient can walk as fast as possible in 6 min on a flat surface, with pauses if necessary. Pulse oxygen saturation, heart rate and Borg index, a rate of perceived exertion, are recorded before and after the test. Along with the total distance walked, the extent of desaturation and heart rate recovery times were linked to the clinical results. The test can also be performed with supplemental oxygen. Absolute contraindications for performing the 6MWT are heart attack or unstable angina in the previous month [71].

### 2.9. Statistical Analysis

Continuous variables were summarized by mean and standard deviation (SD). Continuous variables were assessed by Student’s *t*-test and the Kruskal–Wallis test, whilst the Chi-Square test of independence or Fischer’s exact test were used to compare categorical variables, when appropriate. The analysis of the data was carried out by Software R (version 4.0.2, 2020, R Foundation for Statistical Computing c/o Institute for Statistics and Mathematics Wirtschaftsuniversität Wien Welthandelsplatz 1, 1020 Vienna, Austria). The level of statistical significance was set at a *p*-value of <0.05. The results are given as mean ± SD or median and interquartile range (IQR).

## 3. Results

After the evaluation of all the clinical data of 124 patients, we selected 65 CTEPH patients who met the inclusion criteria (22 male subjects and 43 female subjects; mean age at diagnosis was 69 ± 8). Subjects with uncertain or unconfirmed diagnoses and/or incomplete data were excluded from the statistical analysis and therefore from the study. The diagnosis of CTEPH was validated by finding precapillary pulmonary hypertension on right cardiac catheterization examination, defined as an mPAP ≥ 25 mmHg and a wPAP ≤ 15 mmHg and a PVR > 3 UI, associated with evidence of thromboembolic perfusion defects on lung perfusion scintigraphy and/or chronic thromboembolism on lung angio-CT, after a period of anticoagulant therapy exceeding three months [19,21]. Group excluded: 59/124 patients (28 F-31 M), excluded for evidence under evaluation of idiopathic pulmonary hypertension/associated with an autoimmune disease in 15 patients (Group I), post-pulmonary hypertension capillary in 4 patients (Group II), PH secondary to pulmonary disease in 7 patients (Group III), chronic thromboembolic disease (CTED) in 20 patients, and poor clinical documentation in 13 patients. In total, 26 CTEPH patients were treated with pulmonary endarterectomy (PEA), 32 with medical treatment (MT) and 7 with balloon pulmonary angioplasty (BPA). In the MT group, eleven patients were not eligible for surgery for the finding of distal perfusion defects, nine for high surgical risk and twelve refused surgeries.

In the three groups, there were more females than males (*p* = 0.02 and *p* = 0.04, respectively). The PEA and BPA patients were statistically significantly younger than the MT patients at the time of diagnosis and at enrollment (*p* = 0.04). PEA and BPA patients did not present obesity (*body mass index*, *BMI* ≤ 30), in contrast, 15 patients in the MT group had a BMI ≥ 30 (*p* = 0.03). There was no statistically significant difference at the time of diagnosis between the three groups regarding other clinical characteristics, such as smoking habits and the percentage of various drugs (Table 1).

In the 65 patients, we observed a high percentage of a history of previous pulmonary thromboembolism, reported in 32% of cases, and a history of deep vein thrombosis (DVT) in 61% of cases, the presence of LAC/antiphospholipid antibodies (21%), hypothyroidism in hormone replacement therapy (20%) and the history of neoplastic pathology (31%). The evaluation of risk factors in the three clusters of subjects (Table 2) demonstrated no statistically significant difference (*p* > 0.05).

The analysis of right heart catheterization evaluations, pulmonary function tests, six-minute walk test (6MWT) and blood tests carried out on the three clusters of subjects (Table 3) demonstrated no statistically significant difference at the time of enrollment.

Fortunately, none of our patients treated with PEA had any complications (such as pulmonary reperfusion edema and right-sided heart failure) (48–50). Different drugs (Riociguat: 22 patients, Bosentan (off label): 7 patients, Macitentan (off label): 4 patients, Sildenafil (off label): 2 patients, Tadalafil (off label): 1 patient) for PH administered in the MT subgroup did not show significant differences regarding clinical characteristics and death outcome in our small cohort of the study. Four patients were treated by combination therapy as follows: Riociguat and Bosentan (3 patients) and one by Riociguat and Macitentan. Two PEA patients were treated by Riociguat. One BPA patient received a second-time Riociguat. During the follow-up of three years, no patients in the PEA and BPA groups died; conversely, eleven patients in the MT group died (three patients were treated with combination drugs, four received Riociguat, one patient received Bosentan, another received Macitentan and the last received Sildenafil) (Figure 3).

In the PEA group after 1- and 3-years post-surgery, a significant decrease in plasma BNP values (320 ± 174 vs. 291 ± 183 pg/mL *p*-value = 0.04 and 291 ± 112 vs. 248 ± 105 *p* = 0.03, respectively) and an increase in meters covered at the six-minute walk test (320 ± 62 vs. 336 ± 104 m *p*-value = 0.03 and 326 ± 115 vs. 400 ± 93 m *p* = 0.02, respectively) were observed. There was no statistically significant difference after 1 and 3 years in plasma BNP values or in meters covered at a six-minute walk test in the balloon pulmonary angioplasty group.

## 4. Discussion

The results obtained in this study show that CTEPH is a disease prevalent in old age and most frequent in females, in agreement with data reported in recent literature [24,30]. Furthermore, in agreement with a large prospective international registry of CTEPH patients, in our study population, we observed a high percentage of the history of previous pulmonary thromboembolism and a history of deep vein thrombosis, the presence of LAC/antiphospholipid antibodies, hypothyroidism in hormone replacement therapy and the history of neoplastic pathology [34,35,36,37,38]. All these risk factors confirm their role in favoring the development of chronic pulmonary thromboembolism [34,35,36,37,38]. Some epidemiological studies detected some factors that predispose to the disease, giving some awareness of the etiology of CTEPH [6,7,8,9]. Our data underline that the concomitance of multiple chronic situations, causing a generalized inflammatory condition, may have a role in the development of CTEPH, correlated with the process of fibrinolysis, neoangiogenesis and recanalization of thromboembolic material [34,35,36,37,38]. On the basis of existing data, it can be said that CTEPH develops from a process of remodeling the vessel wall after thromboembolic obstruction, and it is triggered and boosted by the combination of altered angiogenesis, defective fibrinolysis and endothelial impairment [34,35,36,37,38]. This mechanism may be conditioned by predisposing factors, including infections, chronic inflammatory diseases, autoimmune diseases, hypothyroidism in hormone replacement therapies, oncological diseases and thrombophilic plasma factors, and results in the gradual development of PH and, consequently, right heart failure [10,11,12,13,14,15,16,34,35,36,37,38].

The analysis of the clinical symptoms of our patients represents the typical picture of the CTEPH patient. They were frequently characterized by dyspnea, often described as progressively worsening, sometimes associated with other nonspecific symptoms such as atypical chest pain, chest tightness and more rarely with syncopal episodes [2,3,4,35,36,37]. Functional evaluation of patients with CTEPH at the time of diagnosis showed a mild to moderate reduction in DLCO, a reduced 6MWD with associated oxyhemoglobin desaturation and dyspnea of mean intensity of 5 on the Borg scale at the end of the test.

The functional class NYHA at diagnosis resulted more frequently in III-IV and this underlines the literature data; probably due to the long median interval of time often required for diagnosis, approximately 70% of patients with CTEPH showed signs of right heart failure, hypoxemia or hypocapnic respiratory failure and elevated plasma BNP values [2,3,4,69,70,71,72,73]. In accordance with the data reported in the literature, the results of this study confirm the fundamental role of pulmonary endarterectomy (PEA) as the treatment of choice for CTEPH, having demonstrated its effectiveness in significantly improving the clinical, functional and hemodynamic situation in PEA patients [48,49,50]. Within this CTEPH study population, the group of PEA patients reported a statistically significant improvement in the values of BNP and 6MWT after 12 and 36 months of surgery [48,49,50]. Except for a more advanced mean age and a high percentage of obesity compared to the group of operated patients, the inoperable patients did not show statistically significant differences in their clinical, functional and hemodynamic characteristics, in accordance with the international registry data [24,25,26,27,28,29]. Mortality in the PEA and BPA groups was 0%, with good long-term survival. Based on these results, it is possible to underline the importance of each patient with a confirmed diagnosis of CTEPH being promptly evaluated in a highly experienced center for the multidisciplinary assessment of suitability for surgical treatment.

Different drugs for pulmonary hypertension administered in inoperable CTEPH patients did not show significant differences in outcomes in our population. The discrepancy between these data and those reported in the literature can be attributed to the small size of our treatment regime subgroups. In fact, three large, controlled trials (BENEFIT, CHEST and MERIT) showed that medications with an indication for the treatment of PH may generate a certain degree of amelioration on the hemodynamic and functional evaluations and, according to the proof provided, the 2021 ESC/ERS Guidelines warranted their use in patients with CTEPH not suitable for surgery and in subjects with PH not resolved with PEA. Nevertheless, the best use and benefits of many medications are still uncertain, and the poor correlation often resulting between hemodynamic and functional outcomes underlines the need for new real-life studies [2,19,20,21,60,63]. Studies reported in the literature suggest that patients who are inoperable and undergoing medical therapy for PH have a worse prognosis than patients who perform endarterectomy or BPA [24,25,26,27,28,29]. In our group of inoperable CTEPH patients, long-term mortality was 34% within a mean period from diagnosis of 48 months, in accordance with the data published in the literature, in which the three-year survival of subjects treated with medical treatment was 70% [24,25,26,27,28,29]. The group of deceased patients, compared to the group of CTEPH patients who survived under medical therapy, showed the following: rapid progression of the disease, presence of signs of right heart failure, elevated plasma values of BNP (434 pg/mL) and a lower 6MWD (216 m).

## 5. Conclusions

Although this study has some limitations, such as being a single-center study, and the two groups (PEA, MT and BPA) are unbalanced. The major limitation of our study is the small sample size, in particular of the BPA group. It was unfortunate that there was insufficient power to test the differences between BPA patients during the observational period. However, our study population is in line with the literature regarding clinical characteristics and risk factors [7,8,24,25,26]. The percentage of patients not eligible for surgery was 40%. While in the international registry population, the percentage of inoperable patients was 36%. By way of conclusion, CTEPH is a complex disease that demands an individualized approach to therapeutic strategies in centers of expertise, incorporating surgical, interventional, radiological and medical PH expertise with the development of clear outcomes analyses. This study seems to confirm that pulmonary endarterectomy (PEA) can provide an improvement in functional tests in CTEPH. However, future studies are needed to ascertain the causes and relative risk of CTEPH and to confirm the role of PEA and BPA, which appear to be promising treatments for this disease.

## Figures and Tables

**Figure 1 medicina-58-01094-f001:**
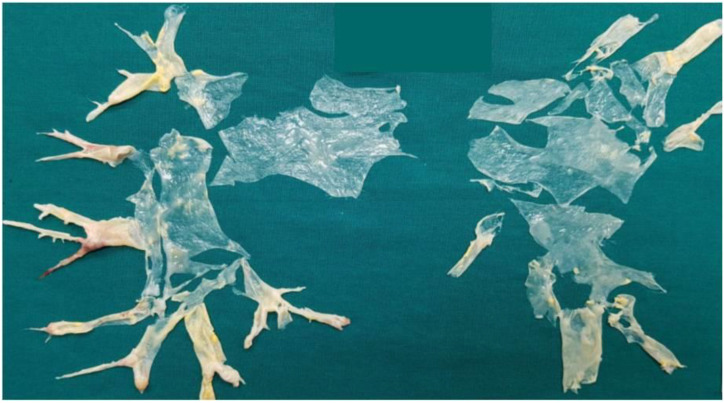
Pulmonary endarterectomy casts from a patient, enrolled in the study, with complex and severe CTEPH.

**Figure 2 medicina-58-01094-f002:**
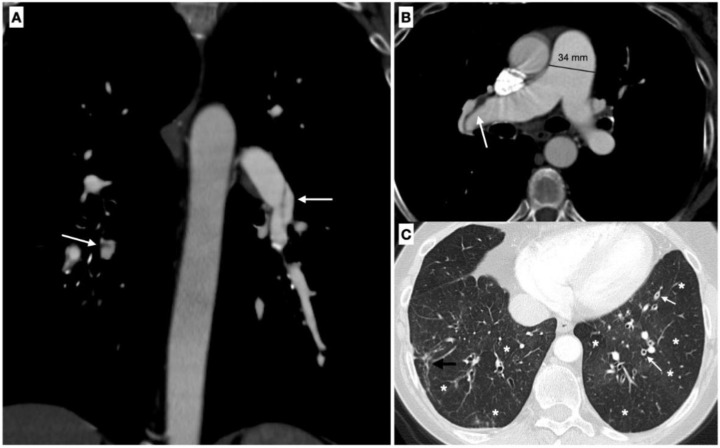
Chronic pulmonary thromboembolism in a 66-y/o woman with a history of episodes of acute PE. (**A**) Coronal image from a pulmonary angio-CT demonstrate the presence of bilateral residual bands within the lumen of the vessel (white arrows), a typical vascular sign of chronic pulmonary thromboembolism. (**B**) Axial image demonstrates the enlargement of the main pulmonary artery (>30 mm), a typical indirect sign of pulmonary hypertension. (**C**) On high-resolution CT can be observed typical parenchymal signs of CTEP, which are represented by a diffuse mosaic perfusion pattern (*), thickening of the bronchial walls (white arrows) and peripheral parenchymal linear opacities from old infarcts (black arrow).

**Figure 3 medicina-58-01094-f003:**
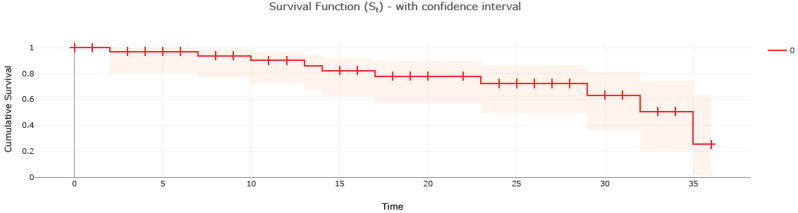
Kaplan–Meier analysis. The survival rate in the medical treatment group within 3 years.

**Table 1 medicina-58-01094-t001:** Characteristics of the study population at enrollment.

Variables	MT	PEA	BPA	*p*-Values
Patients (*N*)	32	26	7	-
Gender (*N*, %)	M: 4 (12%) F 28 (88%)	M: 9 (35%) F: 17 (65%)	M: 2 (29%) F: 5 (71%)	*p* = 0.02 *p* = 0.04
Age at diagnosis Median (IQR)	69 (13)	66 (17)	64 (14)	*p* = 0.04
Age at enrollment Median (IQR)	71 (13)	68 (13)	66 (11)	*p* = 0.03
Current smoker (*N*, %)	15/32 (46%)	11/26 (42%)	3/7 (43%)	*p* = 0.24
Former smoker (*N*, %)	17/32 (53%)	11/26 (42%)	3/7 (43%)	*p* = 0.17
Obesity (*N*, %)	15/32 (46%)	1/26 (11%)	1/7 (14%)	*p* = 0.03
Treatment regimes (PPI/AI/AC)	(28/29/32)	(14/17/26)	(4/5/7)	*p* = 0.16

Legend: *N*: number; % = percentage of patients; M: male; F: female; SD: standard deviation; PPI: proton pump inhibitors; CA: cardioaspirin; AI: antihypertensive drug; AC: anticoagulant therapy; PEA: pulmonary endarterectomy; MT: medical treatment; BPA: balloon pulmonary angioplasty.

**Table 2 medicina-58-01094-t002:** Risk factors of the study population at enrollment.

Risk Factors	MT	PEA	BPA	*p*-Value
**Pulmonary embolism**				
History of PE or DVT	20/32 (62%)	17/26 (65%)	4/7 (58%)	*p* = 0.20
Severe perfusion defect	7/32 (23%)	5/26 (22%)	2/7 (28%)	*p* = 0.18
Chronic pulmonary thromboembolism (angio-CT)	10/32 (30%)	9/26 (33%)	2/7 (28%)	*p* = 0.34
**Thrombophilia**				
LAC/antiphospholipid antibodies	6/32 (20%)	5/26 (22%)	2/7 (28%)	*p* = 0.24
MTHFR mutation	4/32 (12%)	3/26 (11%)	1/7 (14%)	*p* = 0.15
Hyperhomocysteinemia	4/32 (12%)	3/26 (11%)	1/7 (14%)	*p* = 0.64
ATIII, protein C or S defect	4/32 (12%)	3/26 (11%)	1/7 (14%)	*p* = 0.36
Factor II mutation	1/32 (3%)	0/26 (0%)	0/7 (0%)	*p* = 0.67
Factor V mutation	1/32 (3%)	0/26 (0%)	0/7 (0%)	*p* = 0.58
**Clinical conditions**				
Splenectomy	1/32 (3%)	0/26 (0%)	0/7 (0%)	*p* = 0.47
Ventriculoatrial shunting	0/32 (0%)	0/26 (0%)	0/7 (0%)	*p* = 0.54
Infected devices/Intravenous (IV) catheter	0/32 (0%)	0/26 (0%)	0/7 (0%)	*p* = 0.55
Hypothyroidism in replacement therapy	8/32 (25%)	6/26 (23%)	2/7 (28%)	*p* = 0.84
History of cancer	10/32 (30%)	9/26 (33%)	2/7 (28%)	*p* = 0.55

Legend. PE: pulmonary embolism; DVP: deep vein thrombosis; LAC: lupus anticoagulant; PEA: pulmonary endarterectomy; MT: medical treatment; BPA: balloon pulmonary angioplasty.

**Table 3 medicina-58-01094-t003:** Characteristics of the study population at enrollment.

	MT	PEA	BPA	*p*-Value
**Right heart** **catheterization**				
PAPm (mmHg) (Median; IQR)	39 (6)	40 (4)	42 (5)	*p* = 0.47
PVR (dynes/sec/cm-5) (Median; IQR)	539 (94)	557 (85)	525 (78)	*p* = 0.72
PAPs (mmHg) (Median; IQR)	59 (11)	55 (9)	62 (12)	*p* = 0.55
**Spirometry**				
FVC (Median; IQR)	83 (17)	85 (13)	80 (11)	*p* = 0.3
FEV1 (Median; IQR)	95 (18)	92 (12)	96 (13)	*p* = 0.58
IT (Median; IQR)	79 (17)	81 (14)	83 (12)	*p* = 0.67
DLCO (%) (Median; IQR)	60 (12)	63 (25)	59 (14)	*p* = 0.4
**Six-minute walk test (6MWT)**				
Meters (m) (Median; IQR)	298 (54)	310 (76)	305 (82)	*p* = 0.76
Borg scale (Median; IQR)	4 (2)	4 (1)	3 (1)	*p* = 0.38
pO2 (mmHg) (Median; IQR)	64 (23)	67 (41)	65 (33)	*p* = 0.55
**Blood tests**				
BNP (pg/mL) (Median; IQR)	310 (176)	315 (149)	308 (153)	*p* = 0.63

Legend. PEA: pulmonary endarterectomy; MT: medical treatment; BPA: balloon pulmonary angioplasty; mPAP = mean pulmonary arterial pressure, PVR = pulmonary vascular resistance, PAPs = systolic pulmonary pressure, FVC = functional vital capacity, FEV1 = forced expiratory volume in 1 s, IT = Tiffenau index = forced expiratory volume in 1 s/functional vital capacity, DLCO = diffusion capacity of carbon monoxide; PEA: pulmonary endarterectomy; MT: medical treatment; BPA: balloon pulmonary angioplasty.

## Data Availability

All the data are available upon reasonable request to the corresponding author.

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
