# Peer review of "Chronic Thromboembolic Pulmonary Hypertension: An Observational Study"

_medicina, 2022, doi:10.3390/medicina58081094_

Round 1
Reviewer 1 Report
The data analysis was not completed. The difference between any two groups can not be received from the present p value. The conclusion shold be achieved after furher data analysis.
Author Response
R: We would like to thank the reviewer for these comments. In agreement with these observations data analysis was re-evaluated and conclusion rewritten, as reported in the manuscript, in the tables and in the figure.
Reviewer 2 Report
The manuscript entitled " Chronic thromboembolic pulmonary hypertension: Is It Time for a New Therapeutic Era? aims to assess the clinical characteristics of CTEPH patients, surgical or medical treated, in a pulmonology referral center. We have some concerns about this manuscript :
1. Could authors provide the data of subgroup of patient receiving Riociquat ?
2. Any patient receive combine therapy ? BPA+ medication ? PEA + medication ? BPA+PEA ? PEA+BPA+ medication ?
3. The introduction part is too long . Revised might be better
Author Response
The manuscript entitled " Chronic thromboembolic pulmonary hypertension: Is It Time for a New Therapeutic Era? aims to assess the clinical characteristics of CTEPH patients, surgical or medical treated, in a pulmonology referral center. We have some concerns about this manuscript :
- Could authors provide the data of subgroup of patient receiving Riociquat ?
R: We would like to thank the reviewer for comments. In agreement with the reviewer’s observation these sentences have been added: “Different drugs (Riociguat: 22 patiens, Bosentan (off label): 7 patients, Macitentan (off label): 4 patients, Sildenafil (off label): 2 patients, Tadalafil (off label): 1 patients) for PH administered in MT subgroup did not show significant differences regarding clinical characteristics and death in outcome in our small cohort of study. Four patients were treated by combination therapy: Riociguat and Bosentan (3 patients) and one by Riociguat and Macitentan. Two PEA patients after were treated by Riociguat. One BPA patients received in a second time Riociguat. During the follow-up of three years no patients in PEA and BPA groups died, conversely eleven patients in MT group died (three patients treated by combination drugs, four received Riociguat, one patient Bosentan, another Macitentan and the last Sildenafil)” and “Different drugs for pulmonary hypertension administered in inoperable CTEPH patients did not show significant differences in outcome in our population. The discrepancy of these data with those reported in the literature can be attributed to the small size of our treatment regime subgroups. In fact, three large, controlled trials (BENEFIT, CHEST, MERIT) showed that medications with an indication for the treatment of PH may generate a certain degree of amelioration on the haemodynamic and functional evaluations and, according to the proof provided, the 2021 ESC/ERS Guidelines warranted their use in patients with CTEPH not suitable for surgery and in subjects with PH not resolved with PEA. Nevertheless, the best use and benefits of many medications are still uncertain, and the poor correlation often resulted between hemodynamic and functional outcomes underlines the need for new real life studies [2,19-21,60,63]”.
- Any patient receive combine therapy ? BPA+ medication ? PEA + medication ? BPA+PEA ? PEA+BPA+ medication ?
R: We would like to thank the reviewer for comments. In agreement with the reviewer’s observation this sentence has been added: “Fortunately, none of our patients treated with PEA had any complications (such as pulmonary reperfusion edema and right-sided heart failure) (48-50). Different drugs (Riociguat: 22 patiens, Bosentan (off label): 7 patients, Macitentan (off label): 4 patients, Sildenafil (off label): 2 patients, Tadalafil (off label): 1 patients) for PH administered in MT subgroup did not show significant differences regarding clinical characteristics and death in outcome in our small cohort of study. Four patients were treated by combination therapy: Riociguat and Bosentan (3 patients) and one by Riociguat and Macitentan. Two PEA patients after were treated by Riociguat. One BPA patients received in a second time Riociguat. During the follow-up of three years no patients in PEA and BPA groups died, conversely eleven patients in MT group died (three patients treated by combination drugs, four received Riociguat, one patient Bosentan, another Macitentan and the last Sildenafil)”.
- The introduction part is too long . Revised might be better
R: In agreement with the reviewer’s comment the introduction has been revised and abbreviated as reported in the manuscript.
Reviewer 3 Report
This manuscript relates a single center experience in the management of CTEPH. The cohort includes 65 patients that had either PEA, BPA or medical treatment. The main problem I have with this paper is the absence of an explicit working hypothesis: if the idea was to describe the differences between the 3 therapeutic groups, as the procedure was not randomised, the data are just an illustration of some clinical selection criteria that dictated the choice of treatment. If the purpose was to evaluate treatment efficacy, again, as the study was not randomised there is a selection bias with the least favorable cases being in the medical treatment goup. An analysis by propensity score matching or weighing could partially alleviate this bias, but I am afraid that the actual number are to low for such an analysis.
Furthermore the style of the manuscript is an hybrid between a review and a clinical study: the introduction has a far to long paragraph on the etiology and risk factor for CTEPH, and the patients and method section is mixed with general statements that are not necessary for a study report (eg. RHC, angio-CTscan).
The extensive bibliography has to many general reviews.
Specific comments:
1. Were any patients who get more than one therapeutic modiality ? For ex PEA followed by medical treatment ? If yes, how were they counted?
2. It is stated that CTEPH definition and inclusion criterai followed the most recent publications. But one of the reference quoted (Simmoneau 2019) is the one that propose to decrease the mPAP threshold from 25 to 20 mmHg. In this paper the upper limit of normal was 25 mmHg (which is OK) with presumably a PVR > 3 UI ? This has to be clarified in the method section.
3. Table 3 shows data with 2 decimals, which I think is excessive for both the hemodynamics and the PFTs.
4. Table 1 shows a statistical differences in age (both at diagnosis and at enrollment) within the 3 groups. However, looking at the mean and the SD, I have trouble to undertand the p value of 0.03 and 0.04. Statistics mention the use of Mann-Whitney test. However, as there are 3 groups, the Kruskall-Wallis test should be used instead. As these tests are non parametric, the data should be presented as median and range instead of mean and SD.
5. There was no death in the PEA and BPA groups. The survival rate in the medical treatment group is only mentioned as 11 deaths within 3 years in 32 patients. A Kaplan-Meier analysis would give a better view of the outcome for this group
Author Response
This manuscript relates a single center experience in the management of CTEPH. The cohort includes 65 patients that had either PEA, BPA or medical treatment. The main problem I have with this paper is the absence of an explicit working hypothesis: if the idea was to describe the differences between the 3 therapeutic groups, as the procedure was not randomised, the data are just an illustration of some clinical selection criteria that dictated the choice of treatment. If the purpose was to evaluate treatment efficacy, again, as the study was not randomised there is a selection bias with the least favorable cases being in the medical treatment goup. An analysis by propensity score matching or weighing could partially alleviate this bias, but I am afraid that the actual number are to low for such an analysis.
Furthermore the style of the manuscript is an hybrid between a review and a clinical study: the introduction has a far to long paragraph on the etiology and risk factor for CTEPH, and the patients and method section is mixed with general statements that are not necessary for a study report (eg. RHC, angio-CTscan).
The extensive bibliography has to many general reviews.
R: We would like to thank the reviewer for the comments. In agreement with reviewer’s observations we underline in the manuscript the aims of study and the small sample size of our study. Furthermore, several parts have been revised and abbreviated as reported in the manuscript.
Specific comments:
1. Were any patients who get more than one therapeutic modiality ? For ex PEA followed by medical treatment ? If yes, how were they counted?
R: We would like to thank the reviewer for comments. In agreement with the reviewer’s observation these sentences have been added: “Different drugs (Riociguat: 22 patiens, Bosentan (off label): 7 patients, Macitentan (off label): 4 patients, Sildenafil (off label): 2 patients, Tadalafil (off label): 1 patients) for PH administered in MT subgroup did not show significant differences regarding clinical characteristics and death in outcome in our small cohort of study. Four patients were treated by combination therapy: Riociguat and Bosentan (3 patients) and one by Riociguat and Macitentan. Two PEA patients after were treated by Riociguat. One BPA patients received in a second time Riociguat. During the follow-up of three years no patients in PEA and BPA groups died, conversely eleven patients in MT group died (three patients treated by combination drugs, four received Riociguat, one patient Bosentan, another Macitentan and the last Sildenafil)”
- It is stated that CTEPH definition and inclusion criterai followed the most recent publications. But one of the reference quoted (Simmoneau 2019) is the one that propose to decrease the mPAP threshold from 25 to 20 mmHg. In this paper the upper limit of normal was 25 mmHg (which is OK) with presumably a PVR > 3 UI ? This has to be clarified in the method section.
R: In agreement with the reviewer’s observation this value has been added
- Table 3 shows data with 2 decimals, which I think is excessive for both the hemodynamics and the PFTs.
R: In agreement with the reviewer’s observation the data has been corrected
- Table 1 shows a statistical differences in age (both at diagnosis and at enrollment) within the 3 groups. However, looking at the mean and the SD, I have trouble to undertand the p value of 0.03 and 0.04. Statistics mention the use of Mann-Whitney test. However, as there are 3 groups, the Kruskall-Wallis test should be used instead. As these tests are non parametric, the data should be presented as median and range instead of mean and SD.
R: In agreement with the reviewer’s observation the tables have been revised
- There was no death in the PEA and BPA groups. The survival rate in the medical treatment group is only mentioned as 11 deaths within 3 years in 32 patients. A Kaplan-Meier analysis would give a better view of the outcome for this group
R: In agreement with the reviewer’s comment a figure has been added in the manuscript.
Round 2
Reviewer 2 Report
Thank you for revised and response. THe manuscript provide more detail information and become more valuable now.
Author Response
I would like to thank the reviewer for the commentswho helped us improve our manuscript
Reviewer 3 Report
The authors have answered to most of my requests and have amended the text, the tables and the figure, accordingly. In particular, the presentation and statistical analysis of Tables have been corrected. In addition a working hypothesis has been more clearly defined at the end of the introduction.
The present study remains an observational and mostly descriptive study that is confirmatory in nature. For this reason I question the significance of the second part of the title:"Is it time for a new therapeutic era?" As no new management modality is presented, I found this sentence somewhat delusive. In other words, the data presented here do not answer this question
Author Response
I would like to thank the reviewer for all comments that helped us to improve our manuscript.
I agree with the reviewer’s observation that “the present study remains an observational and mostly descriptive study”. For this reason in agreement with the reviewer’s note I changed the title and I deleted the second part.